# Understanding painful versus non-painful dental pain in female and male patients: A transcriptomic analysis of human biopsies

**Biraj Patel[1], Michael A. Eskander[1], Phoebe Fang-Mei Chang[1], Brett Chapa[1], Shivani B. Ruparel[1], Zhao Lai[2,3], Yidong Chen[2,4], Armen Akopian[1], Nikita B. Ruparel[1]***

**1** Department of Endodontics, University of Texas Health Science Center at San Antonio, San Antonio, Texas, United States of America, **2** Greehey Children's Cancer Research Institute, University of Texas Health Science Center at San Antonio, San Antonio, Texas, United States of America, **3** Department of Molecular Medicine, University of Texas Health Science Center at San Antonio, San Antonio, Texas, United States of America, **4** Department of Population Health Sciences, University of Texas Health Science Center at San Antonio, San Antonio, Texas, United States of America

* ruparel@uthscsa.edu

**Data Availability Statement:** Data that support presented findings are available as figures, materials, and methods as well as supplementary

## Abstract

Dental pain from apical periodontitis is an infection induced-orofacial pain condition that presents with diversity in pain phenotypes among patients. While 60% of patients with a full-blown disease present with the hallmark symptom of mechanical allodynia, nearly 40% of patients experience no pain. Furthermore, a sexual dichotomy exists, with females exhibiting lower mechanical thresholds under basal and diseased states. Finally, the prevalence of post-treatment pain refractory to commonly used analgesics ranges from 7–19% ($\sim$ 2 million patients), which warrants a thorough investigation of the cellular changes occurring in different patient cohorts. We, therefore, conducted a transcriptomic assessment of periapical biopsies (peripheral diseased tissue) from patients with persistent apical periodontitis. Surgical biopsies from symptomatic male (SM), asymptomatic male (AM), symptomatic female (SF), and asymptomatic female (AF) patients were collected and processed for bulk RNA sequencing. Using strict selection criteria, our study found several unique differentially regulated genes (DEGs) between symptomatic and asymptomatic patients, as well as novel candidate genes between sexes within the same pain group. Specifically, we found the role of cells of the innate and adaptive immune system in mediating nociception in symptomatic patients and the role of genes involved in tissue homeostasis in potentially inhibiting nociception in asymptomatic patients. Furthermore, sex-related differences appear to be tightly regulated by macrophage activity, its secretome, and/or migration. Collectively, we present, for the first time, a comprehensive assessment of peripherally diseased human tissue after a microbial insult and shed important insights into the regulation of the trigeminal system in female and male patients.

tables. Additionally, all RNA-seq data have been uploaded to NCBI; accession number GSE237398.

**Funding:** This study was funded by NIDCR R03 DE027780-01 (PI: Nikita Ruparel) and seed money from University of Texas Health Science Center San Antonio (PI: Nikita Ruparel). Sequencing data was generated in the Genome Sequencing Facility which is supported by UTHSCSA, NIH-NCI P30 CA054174 (Cancer Center Support Grant at UTHSCSA), NIH Shared Instrument grant S10OD030311(S10 grant), and CPRIT Core Facility Award (RP220662). The funders had no role in study design, data collection and analysis, decision to publish, or preparation of the manuscript.

**Competing interests:** No competing interests

## Introduction

Apical periodontitis is a complex tooth infection-induced inflammatory process [1, 2] that is characterized radiographically by loss of bone in the periapical tissues of the infected tooth, histologically by the formation of granulomatous tissues, and clinically by the presence of mechanical allodynia to chewing (percussion) and/or pushing on the mucosal tissues or skin adjacent to the diseased tooth (palpation). However, despite the known characteristics of apical periodontitis, clinical variability exists in pain phenotypes among patients. Approximately 60% of patients with apical periodontitis present with painful symptoms of mechanical allodynia, while ∼40% present with no symptoms despite the presence of periapical inflammation and bone loss [3, 4]. According to the American Association of Endodontists (AAE) diagnostic terminology, these patients receive a clinical diagnosis of "Asymptomatic Apical Periodontitis". Moreover, male and female patients with apical periodontitis exhibit differences in pain thresholds. Female patients demonstrate a 25% greater reduction in pain thresholds compared to male patients [4, 5]. This is also noted in baseline mechanical thresholds between sexes [5]. Moreover, females have a 4.5 greater odd ratio of persistent pain post-endodontic treatment [root canal treatment] compared to males [6], as well as greater susceptibility to interappointment pain compared to males [7]. These clinical presentations identify two crucial variations 1.] differential mechanisms regulating the trigeminal system in patients diagnosed with apical periodontitis (symptomatic vs. asymptomatic patients), and 2.] nociception-related sexual dimorphism between male and female patients diagnosed with apical periodontitis. The complexity associated with varying clinical phenotypes is likely grounded in the cellular and molecular differences between patients. Furthermore, apical periodontitis is a unique infection-induced clinical condition as antimicrobial drugs provide little-to-no relief in patients with apical periodontitis-induced mechanical allodynia [8–12], suggestive of persistent nociceptive changes that are unaffected by microbial targeting. Therefore, a detailed analysis of peripheral tissues injured upon tooth infection may provide significant insight into pathways and targets that lead to nociceptor activation and inhibition.

The cellular events that mediate tissue destruction and bone resorption in patients with apical periodontitis have largely been attributed to the release of inflammatory cytokines such as prostaglandin E2 (PGE2), tumor necrosis factor-β (TNF-β), IL-1β, interleukin-2 (IL-2) and interleukin-6 (IL-6) [13, 14] in the periapical tissues, secondary to tooth infection. These factors have been identified via discovery research using a fast-performance liquid chromatography gel filtration approach or targeted analyses using enzyme-linked immunosorbent and radioimmuno assays. However, a comprehensive analysis of peripheral tissues from human subjects phenotyped for symptoms and stratified symptoms based on sex has not been investigated. In this study, we, therefore, used a transcriptomic approach to investigate genomic changes in genes that make up periapical pathologic (granulomatous) tissue in patients with apical periodontitis. Specifically, we stratified our data for symptomatic male (SM), asymptomatic male (AM), symptomatic female (SF), and asymptomatic female (AF) patients to comprehensively evaluate differences between patient cohorts.

## Materials and methods

### Ethical approval and informed consent

The collection of human samples was reviewed and approved by the University of Texas Health Science Center at San Antonio Institutional Review Board (IRB) committee (HSC20170318N) and was in accordance with the Declaration of the World Medical Association [www.wma.net]. The IRB committee determined this research as Non-Human Research

as defined by DHHS regulations at 45 CFR 46 and FDA regulations at 21 CFR 56. The proposed project does not include non-routine intervention or interaction with a living individual for the primary purpose of obtaining data regarding the effect of the intervention or interaction, nor do the researchers obtain private, identifiable information about living individuals. All protected health information (PHI) was collected in a de-identified manner by a non-study personnel. Therefore, no informed consent was obtained.

**Study population, periradicular tissue samples, and data collection.** The study population included surgical periapical biopsy samples of patients from the Graduate Endodontics Clinic at University of Texas Health Science Center at San Antonio (UTHSCSA) that were undergoing endodontic microsurgery as part of *standard of care* for treatment of persistent apical periodontitis (chronic disease state). For tissue procurement, all providers were calibrated for clinical diagnosis and tissue excision. Tissue procurement and storage was also standardized for all research personnel. Following an IRB-approved protocol (HSC20170318N) and standardized microsurgical procedures, biopsies of periapical tissues (typically 3–8 mm$^3$) in their entirety were collected along with demographic data between July 2017-August 2020. Pain data for the presence/absence of mechanical allodynia was collected using standardized clinical tests.

*Percussion test*. A tooth with radiographic evidence of periapical pathology and a tooth without radiographic evidence of periapical pathology (control tooth) were gently tapped with the end of a mirror handle along the long axis of the tooth. Responses were recorded as positive (hypersensitive) or negative (within normal limits compared to the control tooth].

*Palpation test*. Firm digital pressure was applied against the facial and lingual/palatal alveolus over the apex/apices of a tooth with radiographic evidence of periapical pathology and a tooth without radiographic evidence of periapical pathology (control tooth). Responses were recorded as positive (hypersensitive) or negative (within normal limits compared to the control tooth).

## Inclusion criteria

1. Patients with a pre-existing indication and consent for endodontic microsurgery after non-healing previous non-surgical endodontic (root canal) treatment

2. Radiographic presence of periapical pathology (loss of bone) on a periapical radiograph for the tooth undergoing microsurgery procedures

## Exclusion criteria

1. Periapical pathology presented as a combined lesion from adjacent teeth as seen on a radiograph

2. Patients with pre-existing chronic pain conditions, such as trigeminal neuralgia and neuropathic pain in the head and neck region

3. Patients taking opioids or chronic immunosuppressant drugs such as steroids, biologicals [for example., Humira®, methotrexate, etc., within the last 90 days prior to surgery

## Definition of groups

**Symptomatic apical periodontitis.**    Teeth with prior endodontic treatment with radiographic signs of periapical pathology where patients exhibit mechanical allodynia to percussion and/or palpation.

**Asymptomatic apical periodontitis.**    Teeth with prior NSRCT with periradicular radiolucencies where patients exhibit <u>no</u> mechanical allodynia to either percussion or palpation test.

## Sample collection

A total of 12 patients who underwent endodontic microsurgery procedures were selected. 6 females (3 symptomatic, 3 asymptomatic) and 6 males (3 symptomatic, 3 asymptomatic). See Table 1 for demographics of the patients included for RNA-seq. At the time of surgery, 3–8 mm$^3$ biopsies were collected in snap-frozen in liquid nitrogen. Samples were then transferred to -80˚C until further use.

## Patient demographics for RNA-seq

Table 1

## RNA extraction and quality assurance

Tissues were homogenized using pre-chilled Omni Bead Ruptor 24 Cryo Cooling Unit (Cat# SKU 19–8005; PerkinElmer, Kennesaw, GA) in 2.8mm ceramic tubes (Cat# 19-628-3; PerkinElmer, Kennesaw, GA) and pre-chilled 700 μl QIAzol Lysis reagent. Total RNA was extracted using the RNAeasy PLUS Universal Mini Kit (Cat# 73404; Qiagen; Hilden, Germany) per the manufacturer's instructions. RNA quality and integrity were checked using Agilent 2100 Bioanalyer RNA 6000 Nano chip (Agilent Technologies, Santa Clara, CA). Only RNA with RIN value of >4 were selected for total RNA sequencing.

## RNA-seq library preparation and next generation sequencing

Approximately 500ng Total RNA was used for RNA-seq library preparation by following the KAPA Stranded RNA-Seq Kit with RiboErase (HMR) sample preparation guide (Cat. No:

**Table 1. "Perc/Palp": Percussion/Palpation.** "+": Patients that responded positively percussion or palpation testing; "-": Patients that responded negatively to percussion or palpation.

| Sample | Age | Race | Symptoms |
| --- | --- | --- | --- |
| | | | Perc/Palp |
| SM#1 | 89 | Hispanic | -/+ |
| SM#2 | 70 | Hispanic | +/+ |
| SM#3 | 62 | Hispanic | +/- |
| AM#1 | 22 | Asian | -/- |
| AM#2 | 82 | White | -/- |
| AM#3 | 50 | White | -/- |
| SF#1 | 49 | Hispanic | -/+ |
| SF#2 | 59 | White | +/- |
| SF#3 | 19 | Hispanic | +/- |
| AF#1 | 72 | Hispanic | -/- |
| AF#2 | 19 | Hispanic | -/- |
| AF#3 | 61 | White | -/- |

KR1151, KAPA Biosystems, USA). The first step in the workflow involved the depletion of rRNA by hybridization of complementary DNA oligonucleotides, followed by treatment with RNase H and DNase to remove rRNA duplexed to DNA and original DNA oligonucleotides, respectively. Following rRNA removal, the mRNA was fragmented into small pieces using divalent cautions under elevated temperature and magnesium. The cleaved RNA fragments were copied into first strand cDNA using reverse transcriptase and random primers. This was followed by second strand cDNA synthesis using DNA Polymerase I and RNase H. Strand specificity was achieved by replacing dTTP with dUTP in the Second Strand Marking Mix (SMM). The incorporation of dUTP in second strand synthesis effectively quenches the second strand during amplification, since the polymerase used in the assay will not incorporate past this nucleotide. These cDNA fragments then went through an end repair process, the addition of a single 'A' base, and then ligation of the adapters. The products were then purified and enriched with PCR to create the final RNA-Seq library. RNA-Seq libraries were subjected to quantification process by the combination of Qubit and Bioanalyzer, pooled for cBot amplification and subsequent sequenced with 50 bp single read sequencing run with Illumina HiSeq 3000 platform. After the sequencing run, demultiplexing with Bcl2fastq2 will be employed to generate the fastq file for each sample. An average of 25-30M reads were obtained for this set of samples.

## Transcriptomic data generation

Sequence reads were aligned to the reference human genome (UCSC human genome build hg19) using TopHat2 [15] with the default setting for stranded library preparation. Then, the obtained BAM files were quantified for read counts at each gene using HTSeq-count [16] and NCBI RefSeq annotation for all samples. Genes located on sex chromosomes were excluded from the differentially expressed gene analysis. The R/Bioconductor package DESeq [17] was used to first normalize gene expression with the size factor method followed by comparisons between groups to identify differentially expressed genes (DEGs) by 1) raw p-value $< 0.05$, 2) at least a 1.5-fold change for up and downregulated genes, and 3) average RPKM (Reads Per Kilobase of gene length per Million reads of the library)$> 1$ in at least one group. Functional analyses of these differentially expressed genes were performed by R/Bioconductor packages topGO (topGO.enw) or function over-representation and fgsea (fgsea.enw) for gene set enrichment analysis. Custom graphics for RNA-seq data, such as volcano plots were generated using R/Bioconductor (https://bioconductor.org/) packages, Venn diagrams using https://bioinfogp.cnb.csic.es/tools/venny/ and heat maps using GraphPad Prism 7.

## Comparison groups for transcriptomic analyses

Symptomatic Males vs. Asymptomatic Males: SM vs. AM
  Symptomatic Females vs. Asymptomatic Females: SF vs. AF
  Symptomatic Females vs. Symptomatic Males: SF vs. SM
  Asymptomatic Females vs. Asymptomatic Males: AF vs. AM

## Determination of biological processes

Biological Processes using PANTHER software did not yield gene clusters relevant to apical periodontitis. Therefore, each candidate gene was manually searched for its function using https://www.genecards.org/ and an in-depth search of the scientific literature as published on https://pubmed.ncbi.nlm.nih.gov/.

## Quantitative Real-Time-PCR [RT-PCR]

Samples were collected using inclusion and exclusion criteria described above. See Table 2 for demographics of the patients included for RT-PCR. Total RNA was isolated as described above. cDNA was prepared using the High Capacity cDNA Reverse Transcription Kit (Cat#4368814; Thermo Fisher Scientific, Waltham, MA). Quantitative real-time PCR (RT-PCR) was performed as previously described [18]. Amplification of target sequences was detected by a sequence detector ABI 7500 Fast RTPCR system (Applied Biosystems, Foster City, CA) using TaqMan Fast Universal PCR Master Mix and CXCL14 (Assay ID Hs01557413_m1; Thermo Fisher Scientific, Waltham, MA) or C3 (Assay ID Hs00163811_m1; Thermo Fisher Scientific, Waltham, MA) or ERAP2 (Assay ID Hs01073631_m1; Thermo Fisher Scientific, Waltham, MA) or CHIT1 (Assay ID Hs00185753_m1; Thermo Fisher Scientific, Waltham, MA) specific primers. The reactions were run in duplicates of 25 μl, including the endogenous control, human GAPDH (Assay ID Hs02786624_g1; Thermo Fisher Scientific, Waltham, MA), for each individual gene expression assay. For quantitative analysis, comparative delta-delta $C_t$ was used to normalize the data based on the endogenous reference and to express it as the relative fold change after the exclusion criteria were verified by comparing primer efficiencies.

## Patient demographics for RT-PCR

Table 2.

**Statistical analyses.**    For qRT-PCR, we used separate patient samples with N = 6-12/group for statistical analyses. Statistical analyses were performed using the Wilcoxon Signed rank test due to the absence of homoscedasticity in our dataset. $P < 0.05$ was considered significant. All data were analyzed using GraphPad Prism 9.

# Results and discussion

## Genes enriched in symptomatic versus asymptomatic patient samples

This dataset aimed to determine genomic differences between symptomatic and asymptomatic patients with pain to mechanical stimulation and patients without, with a clinical diagnosis of apical periodontitis. We first evaluated differentially expressed genes between all male and female patients with pain (symptomatic) and without pain (asymptomatic) using broad criteria of fold change (FC)><1.5 and p-value<0.05. We detected a total of 37 DEGs in SM vs. AM and 79 DEGs in AF vs. AM samples. Further stratification of DEGs yielded 14 genes upregulated in SM compared to AM and 23 genes downregulated in SM compared to AM. Similarly, we found 43 genes upregulated in SF compared to AF and 36 genes downregulated in SF compared to AF. Representative volcano plots for both comparisons are shown in **Fig 1A and 1B**.

Next, we sought to evaluate the presence of unique genes between all SM vs. AM patient samples and SF vs. AF patient samples with RPKM>1. In order to obtain this dataset, we first identified genes unique to all male and female patient samples using criteria of FC><1.5, p-value<0.05, and RPKM>1 (at least in one comparison group). Venn diagram shown in **Fig 1C** identified 37 unique sets of genes expressed *only in male* patients and 77 unique sets of genes expressed *only in female* patients. There were no genes that were expressed in both patient cohort. When each group was further stratified based on symptomatology, we found 23 genes that were upregulated and 14 genes that were downregulated in symptomatic males compared to asymptomatic males (SM vs. AM). Similarly, we found 41 genes that were upregulated and 36 genes that were downregulated in symptomatic females compared to asymptomatic females (SF vs. AF).

**Table 2. "Perc/Palp": Percussion/Palpation.** "+": Patients that responded positively percussion or palpation testing; "-": Patients that responded negatively to percussion or palpation.

| Sample | Age | Race | Symp Perc/Palp | Sample | Age | Race | Symp Perc/Palp | Sample | Age | Race | Symp Perc/Palp |
|--------|-----|------|------|--------|-----|------|------|--------|-----|------|------|
| SM#1 | 69 | Hispanic | +/- | SF#1 | 55 | White | +/+ | AM#1 | 70 | Hispanic | -/- |
| SM#2 | 59 | Hispanic | +/+ | SF#2 | 73 | White | +/+ | AM#2 | 46 | White | -/- |
| SM#3 | 50 | Hispanic | +/- | SF#3 | 72 | Hispanic | +/+ | AM#3 | 54 | White | -/- |
| SM#4 | 19 | White | -/+ | SF#4 | 70 | Hispanic | +/- | AM#4 | 41 | White | -/- |
| SM#5 | 63 | White | +/- | SF#5 | 37 | Hispanic | +/+ | AM#5 | 79 | Hispanic | -/- |
| SM#6 | 68 | Hispanic | +/+ | SF#6 | 64 | White | +/- | AM#6 | 70 | Hispanic | -/- |
| SM#7 | 69 | Hispanic | +/- | SF#7 | 47 | Hispanic | -/+ | AM#7 | 67 | White | -/- |
| SM#8 | 52 | Hispanic | +/- | SF#8 | 65 | White | +/- | AM#8 | 18 | White | -/- |
| | | | | SF#9 | 44 | Hispanic | -/+ | AM#9 | 67 | White | -/- |
| | | | | SF#10 | 39 | Hispanic | -/+ | AM#10 | 67 | White | -/- |
| | | | | SF#11 | 28 | Hispanic | +/- | | | | |
| | | | | SF#12 | 24 | Hispanic | +/- | | | | |

These subsets of genes were then categorized based on their biological functions related to apical periodontitis. Seven relevant functional categories were identified, namely, immune response genes, genes that regulate bone metabolism and repair and regeneration, regulatory genes for vascular and neural functions, and genes for cell adhesion and extracellular matrix remodeling. Any function unrelated to apical periodontitis was classified as "Others". As observed in **Fig 2A–2D**, genes related to immune cell function were predominant in both symptomatic male vs. asymptomatic male (SM vs. AM) patient samples as well as symptomatic female vs. asymptomatic female (SF vs. AF) patient samples. This pattern was also noted when gene selection was restricted to stricter criteria of FC><2.0, p-value<0.05, and RPKM>5 (at least in one comparison group) (**Fig 3A–3D**). Interestingly, genes belonging to this functional class were also seen to be the largest gene set in both upregulated and downregulated genes in all groups. However, each patient subgroup revealed a unique set of genes related to immune cell function, as seen by heatmaps generated for genes within this functional class. **Fig 2E–2H** represent genes with an FC><1.5, p-value<0.05 and RPKM>1 and **Fig 3E–3H** represent genes with an FC><2.0, p-value<0.05 and RPKM>5. Using data from **Fig 3E and 3F**, for all males samples, VNN1 (vascular non-inflammatory molecule-1 or vanin 1), IL10, and GNLY were the top 3 immune response genes upregulated in SM vs. AM (RPKM SM vs. AM: 12.7>4.6, 7.0>1.9 and 7.0>1.0, respectively). Gene VNN1 or vascular non-inflammatory molecule-1 or vanin 1 codes of an enzyme with pantetheinase activity. It is expressed by epithelial cells and perivascular thymic stromal cells. The enzyme acts as a sensor of oxidative-stress [19, 20] as well as mediates thymic stromal cell-assisted T-cell migration to the thymus [19]. While its role in mediating nociception is unknown, it is also well known to induce the release of several known pro-nociceptive inflammatory mediators, namely, IL-6, monocyte chemoattractant protein-1 (MCP-1), and cyclooxygenase-2 (COX-2) [21]. VNN1, therefore, may be a critical regulator of nociception in patients with symptomatic apical periodontitis. GNLY or granulysin is a releasable protein present in cytotoxic T cells and natural killer cells. It functions as an antimicrobial protein by causing microptosis in microbial cells [22]. Additionally, it is also a chemoattractant for various immune cells and induces the expression of multiple inflammatory mediators, one of which is IL-10 [23] which is also upregulated 4 times in SM as compared to AM. Therefore, pathways that activate both GNLY and IL-10 appear to regulate symptoms in patients with apical periodontitis. Interestingly, IL-10 is an extensively studied anti-

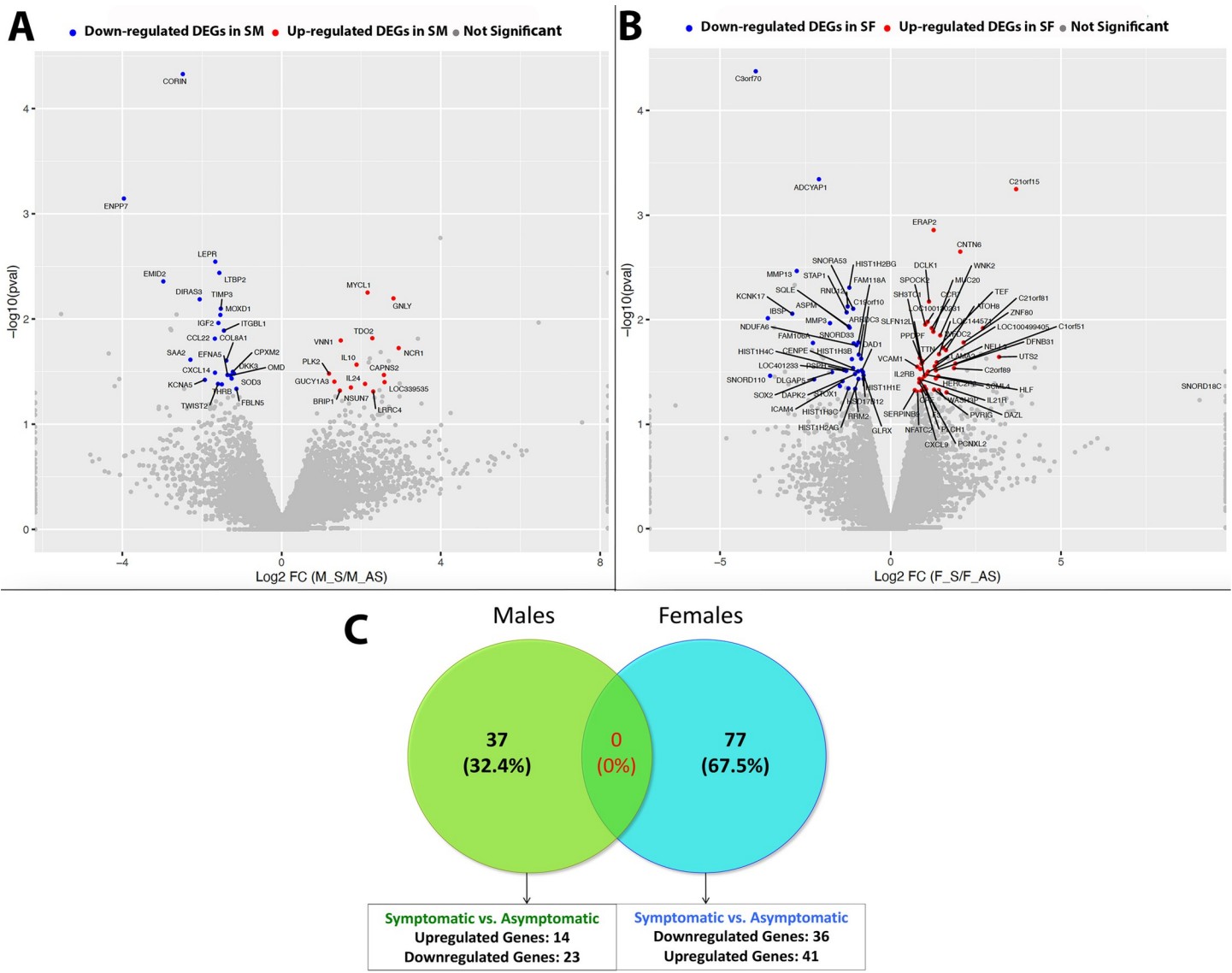

**Fig 1. Bulk RNA sequencing of periapical biopsies from symptomatic and asymptomatic patient samples.** [A] Volcano plot representation of DEGs genes in SM vs. AM using criteria of FC><1.5, p-value<0.05; [B] Volcano plot representation of DEGs in SF vs. AF using criteria of FC><1.5, p-value<0.05; [C] Venn Diagram representation of DEGs in SM vs. AM and SF vs. AF using criteria of FC><1.5, p-value<0.05, and RPKM>1. *FC–Fold Change; RPKM–Reads Per Kilobase of gene length per Million reads of the library.

inflammatory and anti-nociceptive cytokine [24, 25]. This contrary finding may be related to the positive feedback loop associated with the expression of IL-10. Induction of IL-10 has been associated with the induction of pro-inflammatory mediators, and factors that activate pro-inflammatory pathways often overlap with the activation of pathways leading to the production of IL-10 [26]. Moreover, recent work in IL-10 knockout mice demonstrated increased levels of IL-1α in absence of IL-10 and significantly greater infection-induced bone resorption [27]. These findings are suggestive of the co-dependency of IL-10 on pro-inflammatory mediators and may explain the increased expression of IL-10 in symptomatic male patient samples. However, painful symptoms in presence of IL-10 are not fully understood and warrant future functional studies.

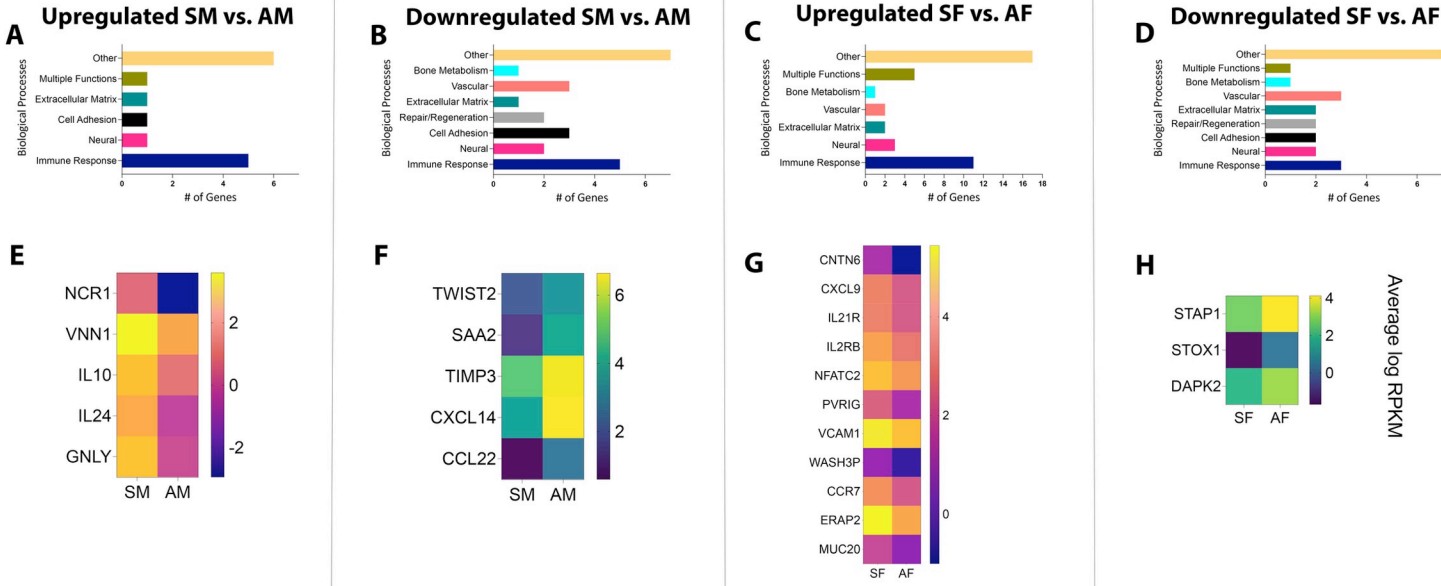

**Fig 2. Biological processes and heat maps of DEGs from symptomatic and asymptomatic patient samples.** Using FC><1.5, p-value<0.05, and RPKM>1, biological processes are presented for genes that were [A] upregulated in SM vs. AM; [B] downregulated in SM vs. AM; [C] upregulated in SF vs. AF; [D] downregulated in SF vs. AF. Using the same criteria, heat maps are presented for DEGs [E] upregulated in SM vs. AM; [F] downregulated in SM vs. AM; [G] upregulated in SF vs. AF; [H] downregulated in SF vs. AF. *FC–Fold Change; RPKM–Reads Per Kilobase of gene length per Million reads of the library.

CXCL14, TIMP3, and SAA2 were the top 3 immune response genes downregulated in SM vs. AM (RPKM SM vs. AM: 33.7<105.4, 33.8<95.9, and 3.8<19.2, respectively]. CXCL14 functions to promote leukocyte migration [28]. However, it is also known to function as an anti-inflammatory and antimicrobial chemokine [28, 29]. Previous work by Sunde et al. demonstrates the presence of bacteria in periapical granulomas [30]. The antimicrobial role of CXCL14 in potentially reducing bacterial toxins that are known sensitizers of nociceptors, such as lipopolysaccharide [31–33], makes it a highly novel target for pain therapeutics in infection-induced pain conditions. TIMP3, or Tissue Inhibitor of Metalloproteinase 3, inhibits the breakdown of extracellular matrix (ECM) proteins and has been studied as a potential

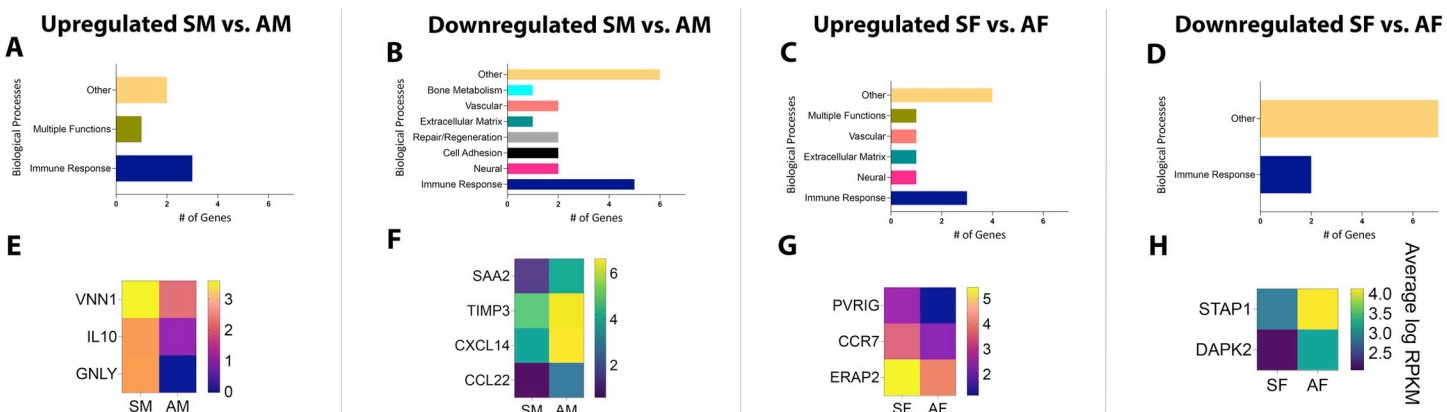

**Fig 3. Biological processes and heat maps of DEGs from symptomatic and asymptomatic patient samples.** Using FC><2.0, p-value<0.05, and RPKM>5, biological processes are presented for genes that were [A] upregulated in SM vs. AM; [B] downregulated in SM vs. AM; [C] upregulated in SF vs. AF; [D] downregulated in SF vs. AF. Using the same criteria, heat maps are presented for DEGs [E] upregulated in SM vs. AM; [F] downregulated in SM vs. AM; [G] upregulated in SF vs. AF; [H] downregulated in SF vs. AF. *FC–Fold Change; RPKM–Reads Per Kilobase of gene length per Million reads of the library.

inhibitor of discogenic pain [34]. Mechanisms of ECM-mediated pain suggest the role of MMP-2 and MMP-9 endopeptidases and the subsequent release of IL-1β in the periphery. However, retrograde uptake of MMP-9 to the spinal cord and the subsequent activation of microglia has also been implicated in the development of neuropathic pain [35, 36]. Moreover, the endogenous inhibitor of ECM, TIMP-1, is a potent blocker of neuropathic pain [35, 36] and maybe a crucial target for pain therapy in apical periodontitis as well. SAA, or serum amyloid A is a primary acute-phase protein that is upregulated under conditions of infection, trauma, and stress, among other environmental insults to a host. While its early role as an acute-phase protein was pro-inflammatory, recent work suggests its role as a protein that functions to maintain homeostasis. This new function is attributed to the tipping of macrophages towards an M2 phenotype and its ability to clear neutrophils from the site of inflammation [37, 38]. The role of M2 macrophages is well-recognized as anti-nociceptive in models of neuropathic pain [39, 40] and musculoskeletal pain [41]. Therefore, the inhibitory role of SAA in asymptomatic apical periodontitis must be evaluated further.

Using data from **Fig 3G and 3H**, for all female samples, ERAP2, CCR7, and PVRIG were the top 3 immune response genes upregulated in SF vs. AF (RPKM SF vs. AF: 42.6>16.9, 11.3>4.3 and 6.1>2.3, respectively). Specifically, ERAP2, or endoplasmic reticulum aminopeptidase 2, is the third candidate gene upregulated in SF compared to AF samples. It is an aminopeptidase involved in trimming antigens into smaller peptides for antigen presentation. Critically, it is involved in downregulating two important anti-inflammatory cytokines, namely, granulocyte-stimulating factor (G-CSF) and IL-10. Both cytokines have been implicated in mediating pain of neuropathic and/or inflammatory origin [24, 25, 42, 43]. CCR7 is a chemokine receptor for chemokines, CCL19, and CCL21, that functions to home subpopulations of T cells and dendritic cells (DC) to regional lymph nodes [44]. Previous work demonstrates the role of peripherally produced CCL21 by primary sensory neurons in inducing centrally mediating tactile allodynia in a model of spinal nerve injury [45]. While this study does not implicate the role of CCR7, infection-induced conditions may likely differ from neuropathic pain and may shed light on the CCL21-CCR7 axis in female patients with symptomatic apical periodontitis. PVRIG or poliovirus receptor-related immunoglobulin domain-containing is a natural killer (NK) cell immune checkpoint receptor and a negative regulator of T cells in cancer patients [46]. Previous work is suggestive of a protective role of NK cells in neuropathy-induced mechanical sensitivity [47]. Therefore, the upregulation of PVRIG in SF warrants further investigation for its association with infection-induced pain in apical periodontitis.

STAP1 and DAPK2 were the top 2 immune response genes downregulated in SF vs. AF (RPKM SF vs. AF: 7.6<17.7, and 4.5<9,1, respectively). STAP1, or signal-transducing adapter protein-1 is a negative regulator of macrophage colony-stimulating factor and macrophage migration [48], and DAPK2 or death-associated protein kinase-2 is involved in programmed cell death [49]. While no direct association of these genes has been evaluated in the pathogenesis of pain, they represent completely novel targets for future functional studies and will be crucial in deducing pathways and cell types that inhibit microorganism-induced pain states.

Genes pertaining to other biological processes are shown in **S1–S4 Tables**.

## Genes enriched in male versus female patient samples

This dataset aimed to determine genomic differences between male and female patients with pain to mechanical stimulation and male and female patients without pain to mechanical stimulation, with a clinical diagnosis of apical periodontitis. We first evaluated differentially expressed genes (DEGs) between all male patients with (symptomatic) and without pain

(asymptomatic) and female patients with (symptomatic) and without pain (asymptomatic) using broad criteria of fold change (FC)><1.5 and p-value<0.05. We detected a total of 53 DEGs in SF vs. SM and 54 DEGs in AF vs. AM samples. Further stratification of DEGs yielded 26 genes upregulated in SF compared to SM and 28 genes downregulated in SF compared to SM. Similarly, we found 27 genes upregulated in AF compared to AM and 26 genes downregulated in AF compared to AM. Representative volcano plots for both comparisons are shown in **Fig 4A and 4B**.

Next, we sought to evaluate the presence of unique genes between all SF vs. SM patient samples and AF vs. AM patient samples with RPKM>1. In order to obtain this dataset, we first identified genes unique to all symptomatic and asymptomatic patient samples using criteria of FC><1.5, p-value<0.05, and RPKM>1 (at least in one comparison group). Venn diagram shown in **Fig 4C** identified 51 unique sets of genes expressed *only in symptomatic* patients and 50 unique sets of genes expressed *only in asymptomatic* patients. There were 3 genes that were expressed in both patient cohorts. When each group was further stratified based on symptomatology, we found 26 genes that were upregulated and 29 genes that were downregulated in symptomatic females compared to symptomatic males (SF vs. SM). Similarly, we found 27 genes that were upregulated and 26 genes that were downregulated in asymptomatic females compared to asymptomatic males (AF vs. AM).

These subsets of genes were then categorized based on their biological functions related to apical periodontitis. The same seven functional categories were identified, as seen in **Fig 5A–5D**. Genes with multiple functions were named "Multiple functions". As observed in **Fig 5A–5D**, genes related to immune cell function were predominant in both symptomatic female vs. symptomatic male (SF vs. SM) patient samples as well as asymptomatic female vs. asymptomatic male (AF vs. AM) patient samples. This pattern was also noted when gene selection was restricted to stricter criteria of FC><2.0, p-value<0.05, and RPKM>5 (at least in one comparison group) (**Fig 6A–6D**). Immune response genes were the largest functional category in both upregulated and downregulated genes in all groups. However, each patient subgroup revealed a unique set of genes related to immune cell function, as seen by heatmaps generated for genes within this functional class. **Fig 5E–5H** represent genes with an FC><1.5, p-value<0.05 and RPKM>1 and **Fig 6E–6H** represent genes with an FC><2.0, p-value<0.05 and RPKM>5. Using data from **Fig 6E and 6F**, for all symptomatic samples, C3, ERAP2, and CHIT1 were the top 3 genes upregulated in SF vs. SM (RPKM SF vs. SM: 157.4>59.6, 42.6>14.3 and 18.6>1.8, respectively). C3 or complement 3 plays a critical protein required for complement activation and therefore is a member of the innate immune response. The complement system has been implicated in several pain states. Specifically, the C3 protein has been strongly associated with multiple preclinical pain models such as sciatic nerve ligation [50–52], spinal inflammatory neuropathy [53, 54], incisional injuries [55], spinal cord injury [56], Complete Freund's Adjuvant (CFA) [57], and direct injection of C3 in rodent paw [58, 59]. Additionally, increased C3 levels have been noted in patients with rheumatoid arthritis, sickle cell anemia, and patients with groin pain [60–62]. However, a far more complex relationship exists between C3 levels and disease predisposition between males and females. Healthy female volunteers have lower levels of C3 compared to men [63]. Still, lower C3 levels have been associated with a greater risk for systemic lupus erythematosus (SLE) and Sjögren's syndrome in women versus men [64]. Collectively, these and our own findings make C3 a highly important target for female patients with symptomatic apical periodontitis. As mentioned above, ERAP2 was also upregulated in SF compared to AF samples and appears to be a critical regulator of nociception in female patients by downregulating anti-inflammatory cytokines, namely, granulocyte-stimulating factor (G-CSF) and IL-10. It has been hypothesized that ERAP2 may play a compensatory role in females [65] and that sexual dichotomy does exist, with functional differences

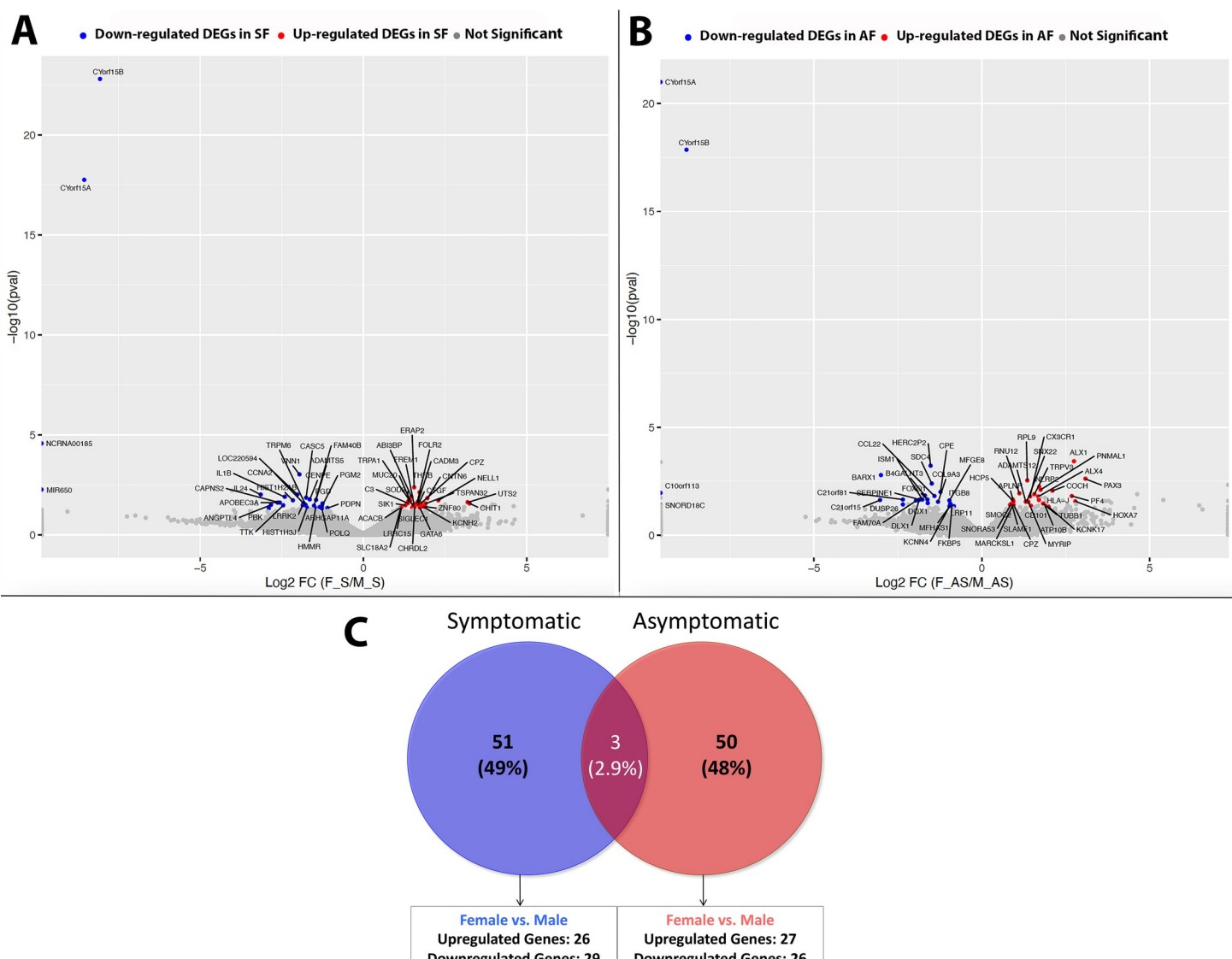

**Fig 4. Bulk RNA sequencing of periapical biopsies from female and male patient samples.** [A] Volcano plot representation of DEGs genes in SF vs. SM using criteria of FC><1.5, p-value<0.05; [B] Volcano plot representation of DEGs in AF vs. AM using criteria of FC><1.5, p-value<0.05; [C] Venn Diagram representation of DEGs in SF vs. SM and AF vs. AM using criteria of FC><1.5, p-value<0.05 and RPKM>1. *FC–Fold Change; RPKM–Reads Per Kilobase of gene length per Million reads of the library.

noted between males and females. Additionally, mice do not express ERAP2 gene [66]; this further signifies the identification of ERAP2 in our study from a translational therapeutic perspective. CHIT1 or chitotriosidase or chitinase is a second candidate gene expressed at greater levels in SF vs. SM. CHIT1 is secreted by activated macrophages and plays a role in the degradation of chitin-containing pathogens. There has been no prior evidence for the role of CHIT1 in pain or to have a female predilection, and therefore presents as a novel target for pain associated in female patients with apical periodontitis.

IL-1β, PDPN (podoplanin), and VINN1 were the top 3 and only genes downregulated in SF vs. SM (RPKM SM vs. AM: 8.8<44.0, 11.0<22.6, and 12.7<3.4 and respectively). There is extensive literature demonstrating the pro-nociceptive role of IL-1β in the periphery and as

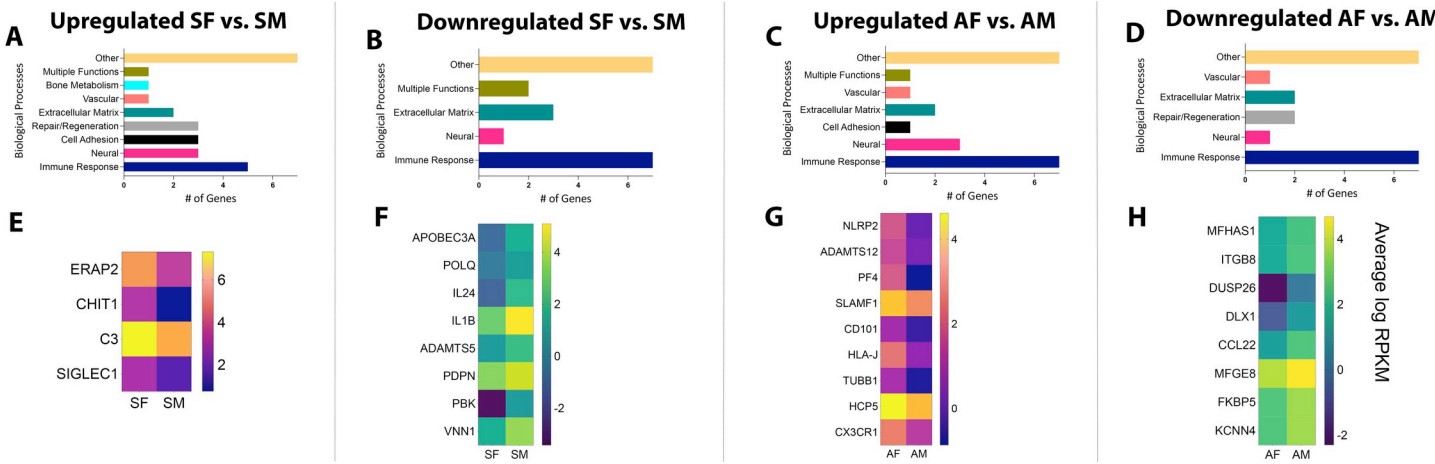

**Fig 5. Biological processes and heat maps of DEGs from female and male patient samples.** Using FC><1.5, p-value<0.05, and RPKM>1, biological processes are presented for genes that were [A] upregulated in SF vs. SM; [B] downregulated in SF vs. SM; [C] upregulated in AF vs. AM; [D] downregulated in AF vs. AM. Using the same criteria, heat maps are presented for DEGs [E] upregulated in SF vs. SM; [F] downregulated in SF vs. SM; [G] upregulated in AF vs. AM; [H] downregulated in AF vs. AM. *FC–Fold Change; RPKM–Reads Per Kilobase of gene length per Million reads of the library.

well as in the spinal system [67]. Data also exists for its role in regulating glial-neuronal interactions in neuropathic pain states, as well as the role of the inflammasome in regulating this pro-inflammatory cytokine [67]. Evidence of expression of IL-1β has been demonstrated in periapical lesions with increasing levels positively associated with the severity of inflammation [13, 68, 69]. Contrary to our findings in the peripheral tissue of periapical biopsies, prior literature suggests 5-28-fold great secretion of IL-1β from mononuclear cells in circulation from women than men [70]. However, other reports show higher expression of the gene in men with Kawasaki disease than in women [71] and that polymorphisms in the gene affect depression-like symptoms in men more than in women [72]. Symptomatic apical periodontitis in men may therefore represent a subgroup that is tightly regulated by IL-1β and a cytokine that may be targeted for future therapeutics. PDPN, or podoplanin, is a well-established marker of lymphatic endothelial cells but has recently been discovered on subsets of lymphocytes, follicular dendritic cells, and macrophages [50]. It functions to mobilize dendritic cells to regional

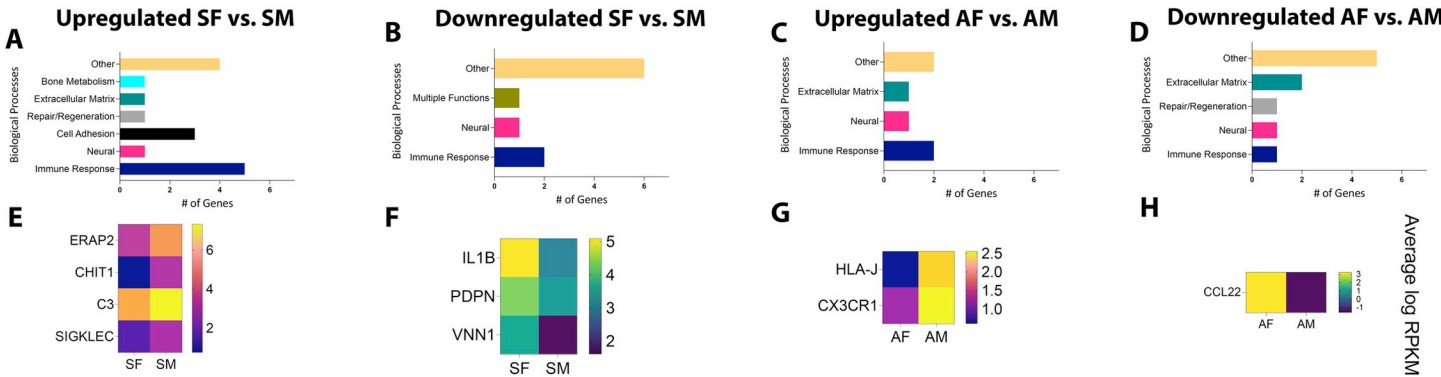

**Fig 6. Biological processes and heat maps of DEGs from female and male patient samples.** Using FC><2.0, p-value<0.05, and RPKM>5, biological processes are presented for genes that were [A] upregulated in SF vs. SM; [B] downregulated in SF vs. SM; [C] upregulated in AF vs. AM; [D] downregulated in AF vs. AM. Using the same criteria, heat maps are presented for DEGs [E] upregulated in SF vs. SM; [F] downregulated in SF vs. SM; [G] upregulated in AF vs. AM; [H] downregulated in AF vs. AM. *FC–Fold Change; RPKM–Reads Per Kilobase of gene length per Million reads of the library.

lymph nodes, increase phagocytic activity of macrophages and exacerbate inflammatory disease progression [50]. Despite no prior association with the pathophysiology of nociception, PDPN activity in cells of the immune response may play an important role in the nociceptive response observed in SM. VNN1, as mentioned above, is expressed in greater amounts in SM compared to AM and appears to also regulate sex-specific pain states in males versus females (SM vs. SF). VNN1, therefore, is novel target for investigation for male patients with symptomatic apical periodontitis.

Using data from **Fig 6G and 6H**, for all asymptomatic samples, CX3CR1 and HLA-J were the only top 2 genes upregulated in AF vs. AM (RPKM AF vs. AM: 6.4>2.3 and 5.1>1.5, respectively). CX3CR1 or C-X3-C motif chemokine receptor 1. It is a chemokine receptor for a chemokine, fractalkine, also known as CX3CL1. CX3CR1 is therefore known as the fractalkine receptor or G-protein coupled receptor 13 (GRP13). There is a vast body of literature that suggests a pro-nociceptive role of microglial CX3CR1 in spinal and supraspinal sites in chronic pain conditions [73, 74]. This is contrary to the association found in our study. Our findings demonstrate the upregulation of CX3CR1 in AF with an RPKM of 6.4 vs. AM with an RPKM of 2.3. Previous work also suggests differential functional effects of peripheral versus spinal fractalkine activity. In a model of sciatic nerve injury, Holmes and colleagues [75] demonstrate delayed development of thermal and mechanical allodynia after peripheral intra-neural administration of fractalkine and that CX3CR1 knockout mice showed increased allodynia in mice. The role of CX3CR1 appears to be in resident macrophages in the dorsal root ganglion and/or spinal microglia; however, the role of sensory neuronal CX3CR1 cannot be discounted [76]. Sex-related differences in CX3CR1 functions reveal a protective role of the receptor in females versus males [77]. Therefore, the role of CX3CR1 in the nociceptive pathway appears to be pleiotropic. Evaluating the peripheral role of CX3CR1 in apical periodontitis in both females and males would shed critical therapeutic insights into the receptor as a druggable target. HLA-J, or Major Histocompatibility Complex, Class I, J (Pseudogene), has been long known to be a pseudogene due to imperfections in the gene with no functional activity associated with it. However, a recent study reveals transcriptional activities with a functional effect in breast cancer patients [78]. To our knowledge, no other evidence for a functional role of HLA-J exists, and our finding that HLA-J is one of only two immune response genes found to be upregulated in AF patients with an RPKM>5 points to a previously unknown role of this gene in apical periodontitis-mediated nociception. HLA-G shares high sequence homology with HLA-J [78], which has been widely known as an immunosuppressive protein [78]. This may provide a starting point for deducing the functional role of HLA-J in infection-induced antinociception, especially in females.

CCL22 was the only gene downregulated in AF vs. AM (RPKM AF vs. AM: 2.3<6.2). CCL22 has recently been implicated as a pro-nociceptive chemokine by direct activation of sensory neurons in a model of post-surgical pain [79]. However, it is also known to be part of the M2a macrophage secretome [80] and affects chemotaxis of T regulatory cells (Tregs) that are immunosuppressive [81]. Previous work using an anti-CD25 antibody that depletes Tregs, demonstrated prolonged mechanical pain sensitivity in a model of partial sciatic nerve ligation [82]. Our data demonstrate significant upregulation of CCL22 in AM as compared to SM (Fig 3F) as well as AF (Fig 6H). The role of CCL22 in the inhibition of apical periodontitis-induced pain in male patients may be linked to the increased number of Tregs in periapical biopsies and warrants future functional analyses.

Genes pertaining to other biological processes are shown in **S5–S8 Tables**.

## Validation of genes differentially regulated in patients groups

Using biological samples (N = 6–12) entirely different from the ones used in generating RNA sequencing datasets, we conducted validation studies using RT-PCR. Only genes with criteria of FC><2.0, p-value<0.05, and RPKM>5 were selected for validation purposes. Our data demonstrate that CXCL14 with RPKM values of 33.7<105.4 for SM < AM was seen to be upregulated by 4.3-fold in AM compared to SM (**Fig 7A**). This aligned with sequencing data demonstrating a 3.1-fold change in AM compared to SM samples. C3 with RPKM values of 157.4>59.6 for SF > SM showed upregulation by 3.3-fold in SF compared to SM samples (**Fig 7B**). Its corresponding fold change from RNA sequencing was 2.5. Other genes, namely, ERAP2 with an RPKM of 42.6>14.3 and CHIT1 with RPKM values of 18.6>1.8 for SF vs. SM, showed a 1.75-fold and 1.9-fold upregulation, respectively in SF compared to SM samples (**Fig 7C and 7D**). Their respective fold changes, as seen in RNA sequencing data, show a 2.5-fold and 9.5-fold upregulation in SF compared to SM samples.

## Conclusions

Diversity in clinical pain phenotypes raises important questions about the differential regulation of the nociceptive pathway in humans. Examples of such diversity exist in chemotherapy-induced neuropathy patients, oral cancer patients, and patients with dental infections. Each of the aforementioned conditions have a subset of patients that do not experience any pain despite the development of an advanced pathologic disease state [3, 4, 83–85]. Apical periodontitis, a sequela of tooth infection, affects ∼20% of patients worldwide [86, 87]. Despite this, the lack of presence of pain in 40% of the patient population poses a significant health concern to the prevalence of untreated apical periodontitis. However, this cohort represents patients with mechanisms that shed insight into inhibitory pain pathways for future pharmacotherapeutics. In this study, using a transcriptomic approach, we sought to investigate genomic changes within peripheral tissues (periapical biopsies) affected by apical periodontitis from human subjects. Specifically, we stratified our data based on symptoms (symptomatic vs. asymptomatic) and sex (female vs. male) to comprehensively evaluate differences between symptomatic and asymptomatic patients, as well as sex-related differences regulating clinical pain phenotype in patients.

Using bulk RNA sequencing of periapical biopsies, our data identified several uniquely expressed genes that regulate immune cell functions between symptomatic and asymptomatic patients within both female and male cohorts (SM vs. AM and SF vs. AF). Additionally, genes of the immune response pathways also appear to tightly regulate nociception or inhibition of nociception between sexes (SF vs. SM and AF vs. AM). This is conceivable as periapical granulomatous tissues are predominately made up of myeloid cells and lymphocytes, namely T and B cells, mast cells, among other non-prominent non-immune cells, such as sensory neuronal terminals innervating the inflamed periapical tissue [88–91]. However, genes that regulate nociception versus inhibition of nociception in males (SM vs. AM) and females (SF vs. AF) are entirely different. Similarly, genes that regulate sex-related differences within all symptomatic and asymptomatic samples were found to be distinctive with no overlap between sexes. This sheds critical insights into the pathways regulating nociception in both sexes. Overall, genes regulating pain in symptomatic patients appear to primarily modulate functional effects in T cells and NK cells that represent the active phase of an immune response. On the other hand, in asymptomatic patients, modulation of M1/M2 macrophage phenotype, as well as genes involved in tissue homeostasis, appear to represent biological processes that contribute to wound healing and resolution. To this end, macrophage polarization has been strongly implicated in the severity of apical periodontitis [49, 92]. Furthermore, M1/M2 macrophage ratio

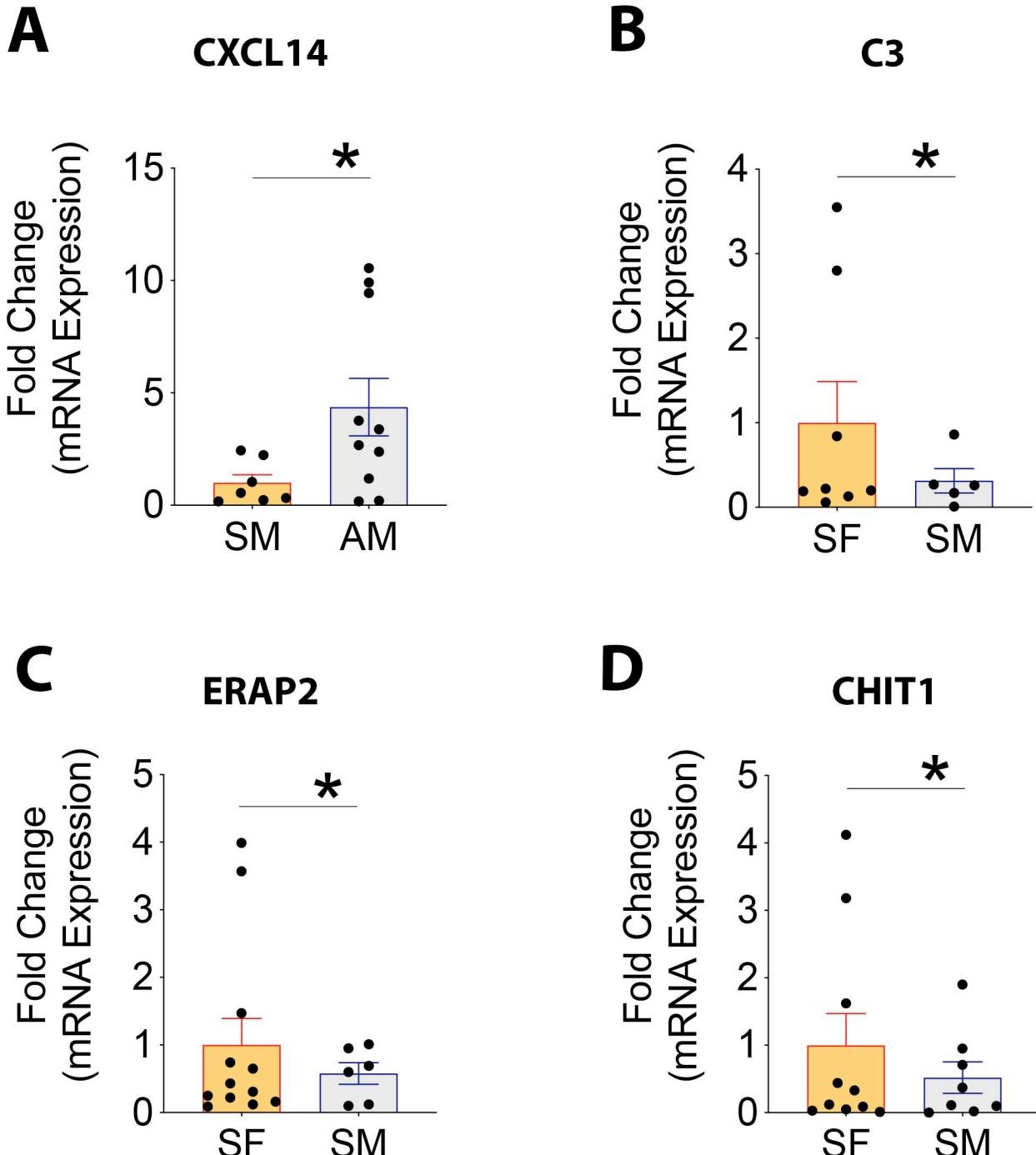

**Fig 7. Quantitative RT-PCR validation of DEGs identified by bulk RNA-sequencing.** [A] Expression of CXCL14 in samples from SM and AM patients; [B] Expression of C3; [C] ERAP2; and [D] CHIT1 in samples from SF and SM patients. Data is normalized to SM for CXCL14 and SF for C3, ERAP2, and CHIT1. Data were analyzed using Wilcoxon Signed Rank Test. *p<0.01.

was greater in symptomatic patient samples compared to asymptomatic samples [49]. Such differences have been previously postulated to explain the radiographic and clinical differences seen between symptomatic and asymptomatic apical periodontitis [93] and shed greater insight into pain mechanisms between the two patient cohorts. Similarly, sexual dimorphism

noted in patients with and without symptoms also greatly appears to be regulated by biological processes governing the innate immune response. More specifically, the majority of the genes discussed here regulate macrophage function as seen by genes that modulate the migratory activity of macrophages, the secretome of macrophages, or the direct activation of macrophages via ligands such as fractalkine for an antinociceptive response. Given that granulomatous tissues are primarily a congregation of macrophages, it is not surprising that this cell type is crucial for the differences observed between female and male patients.

Persistent pain from a microbial infection posits the question of the diversity of microbes as a variable between symptomatic and asymptomatic patients. To this end, several microbial species have been investigated and have been shown to be associated with symptoms, including black-pigmented bacteroides species [94–96], fusobacterium species [97], and lactobacillus species [98, 99]. Some studies suggest that greater diversity exists in symptomatic versus asymptomatic cases [100–102]. However, more recently, the "Microbial Community" concept, rather than a single-species culprit, has been proposed to explain the differences observed in the pathophysiology of symptomatic and asymptomatic apical periodontitis. In this concept, the presence or absence of symptoms is determined not only by the density and virulence factors of the different bacterial species present but also by the host's immune response [103, 104]. While we did not determine the microbial diversity in our patient population, it is likely that each patient had a distinct microbial "community" that differed from the other patients within the same group. Despite this assumption, our findings identified host immune response genes to be the predominant subset of genes that demonstrated the greatest differential regulation between symptomatic and asymptomatic genes. This was also true between female and male patient samples and highlights the crucial role of the immune response in regulating both symptoms as well as sex-related pain phenotypes.

## Limitations

We recognize some shortcomings of our study. 1.) We conducted bulk RNA sequencing to determine genomic changes between patient cohorts as well as assess the average expression level of target genes across all populations of cells within periapical biopsy samples. This precludes the identification of cell types that express the target genes as well as cell-to-cell variability in expression levels. While single-cell RNA sequencing would circumvent some of the drawbacks of bulk RNA sequencing, the inability to isolate primary sensory neurons from peripheral tissues, as well as cell isolation of any rare population of cells, poses a considerable challenge of this approach to the overall aim of our study. 2.) A small sample size of N = 3 with a wide age range and race was used for bulk RNA sequencing, which potentially reduces the generalizability of our findings. Moreover, the majority of patients declined a histopathologic assessment of the biopsy samples. Given the incidence of periapical granuloma vs. periapical abscess vs. periapical cyst [105–108], the differential cellular compositions of each of these diagnoses present a biological confound in our patient samples. However, we conducted validation studies using RT-PCR of several target genes with separate biological samples with an N = 6–10 per group. This greatly increases the rigor and internal validity of our findings and minimize the impact of age, race or histopathology-related variability. 3.) In lieu of the challenges faced in isolating human trigeminal ganglia, as well as cost considerations in acquiring human trigeminal tissue, we used peripheral pathologic biopsies for our transcriptomic analyses. The use of peripheral human tissue, however, provides unique translatability as it captures functional (nerve sprouting, immune cell infiltration, increased vascularity] and regulatory changes within cell types under a diseased state. This is a distinctive advantage over the use of naïve post-mortal tissues. Moreover, primary sensory neurons likely represent less than 1% of

all cells within the trigeminal ganglia innervating a single tooth which would further cause technical limitations in sequencing trigeminal neurons. Finally, preclinical data strongly suggests tissue-specific differences in innervation. Therefore, the transcriptomic study of microbial-induced insult of periapical tissues surrounding teeth offers entirely novel advantages previously unknown in apical periodontitis.

Collectively, our study, for the first time, comprehensively identified genes and biological processes that regulate symptoms and sex-related differences in pain phenotypes in patients with apical periodontitis. The translational significance of this study is underscored by the use of diseased human peripheral tissue biopsies for the generation of the meta-transcriptomic dataset, as well as validation studies with separate patient samples that greatly increase the rigor and internal validity of the data. Additionally, the use of human tissues stratified for females and males greatly advances our knowledge of the female predilection for frequency and severity of pain in other persistent pain conditions [109]. The identification of genes that function in host immunity, resolution, and homeostasis paves the way for future functional studies for pain therapeutics.

## Supporting information

**S1 Table. Genes upregulated in symptomatic males compared to asymptomatic males.** (DOCX)

**S2 Table. Genes downregulated in symptomatic males compared to asymptomatic males.** (DOCX)

**S3 Table. Genes upregulated in symptomatic females compared to asymptomatic females.** (DOCX)

**S4 Table. Genes downregulated in symptomatic females compared to asymptomatic females.** (DOCX)

**S5 Table. Genes upregulated in symptomatic females compared to symptomatic males.** (DOCX)

**S6 Table. Genes downregulated in symptomatic females compared to symptomatic males.** (DOCX)

**S7 Table. Genes upregulated in asymptomatic females compared to asymptomatic males.** (DOCX)

**S8 Table. Genes downregulated in asymptomatic females compared to asymptomatic males.** (DOCX)

## Author Contributions

**Conceptualization:** Nikita B. Ruparel.

**Data curation:** Biraj Patel, Shivani B. Ruparel, Yidong Chen, Nikita B. Ruparel.

**Formal analysis:** Shivani B. Ruparel, Yidong Chen, Armen Akopian, Nikita B. Ruparel.

**Investigation:** Nikita B. Ruparel.

**Methodology:** Michael A. Eskander, Phoebe Fang-Mei Chang, Brett Chapa, Shivani B. Ruparel, Zhao Lai, Yidong Chen, Nikita B. Ruparel.

**Project administration:** Nikita B. Ruparel.

**Resources:** Nikita B. Ruparel.

**Supervision:** Nikita B. Ruparel.

**Validation:** Michael A. Eskander, Nikita B. Ruparel.

**Writing – original draft:** Biraj Patel, Nikita B. Ruparel.

**Writing – review & editing:** Nikita B. Ruparel.

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
