## [Decision Letter · Decision Letter 0]

27 Jun 2023

PONE-D-23-12043Understanding Painful versus Non-Painful Dental Pain in Female and Male Patients: A Transcriptomic Analysis of Human BiopsiesPLOS ONE

Dear Dr. Ruparel,

Thank you for submitting your manuscript to PLOS ONE. After careful consideration, we feel that it has merit but does not fully meet PLOS ONE’s publication criteria as it currently stands. Therefore, we invite you to submit a revised version of the manuscript that addresses the points raised during the review process.

We look forward to receiving your revised manuscript.

Kind regards,

Artak Heboyan, Ph.D.

Academic Editor

PLOS ONE

Journal Requirements:

“None to disclose”

6. Please upload a new copy of Figures 1 and 4 as the detail is not clear. Please follow the link for more information: https://blogs.plos.org/plos/2019/06/looking-good-tips-for-creating-your-plos-figures-graphics/

7. We note you have included a table to which you do not refer in the text of your manuscript. Please ensure that you refer to Table 1 in your text; if accepted, production will need this reference to link the reader to the Table.

Reviewers' comments:

Reviewer's Responses to Questions

**Comments to the Author**

1. Is the manuscript technically sound, and do the data support the conclusions?

Reviewer #1: Partly

Reviewer #2: Yes

2. Has the statistical analysis been performed appropriately and rigorously? 

Reviewer #1: Yes

Reviewer #2: Yes

3. Have the authors made all data underlying the findings in their manuscript fully available?

Reviewer #1: Yes

Reviewer #2: Yes

4. Is the manuscript presented in an intelligible fashion and written in standard English?

Reviewer #1: Yes

Reviewer #2: Yes

5. Review Comments to the Author

Reviewer #1: This manuscript mainly reports the RNA-seq results of the periapical tissue of patients with periapical periodontitis, compares the gene expression differences between symptomatic and asymptomatic, or male and female, and uses RT-qPCR to verify key genes. There are however some concerns that need to be addressed satisfactorily by the authors before this manuscript could be considered acceptable.

1. About the Exclusion and Inclusion Criteria:

1) 12 patients were included for RNA-seq, however, I notice that the the oldest was 89 years old and the youngest was 19 years old. I wonder if age is a factor that affects the RNA-seq results? The same concern applies to the race of the patients who were included.

2) There is a lack of description of the details of each patient's condition, such as the degree and scope of periapical resorption in each patient, and whether the difference in the degree of infection will also affect the RNA-seq results?

2. About the sample collection:

The author mentions “3-8 mm3 biopsies were collected” in the manuscript, but lacks the description of details, for example, were the biopsies taken from the apical foramen or from the center of the lesion? Or randomly take 3-8mm3 from the diseased tissue removed during the surgery?

3. About the RT-PCR:

1) In the section of Results, the authors mention that “Using biological samples (N=6-12) entirely different from the ones used in generating RNA sequencing datasets, we conducted validation studies using RT-PCR.” In the Methods section the authors do not mention these details.

2) Do the samples for PCR meet the same exclusion and inclusion criteria? I think the patient information for PCR should also be listed as a table in the methods section.

4. About the Symptomatic or Asymptomatic patient:

1) The authors define whether symptoms are present through palpation and percussion, and patients with mechanical allodynia to palpation or percussion are considered symptomatic patients. Table 1 should also indicate whether patients with mechanical allodynia to both examinations or only one examination.

2) Were all examinations performed by the same person, and are the standards the same? The authors should add the description of the details of the examination.

Reviewer #2: This is a well conducted study, which is likely to have a great impact once published.

there are some minor comments: Fig 1-A and 1B and Fig 4: legends are not visible, please enlarge/bold for further clarity for the reader.

Please make sure all the abbreviations are defined at their first appearance in the manuscript.

6. PLOS authors have the option to publish the peer review history of their article (what does this mean?). If published, this will include your full peer review and any attached files.

Reviewer #1: No

Reviewer #2: No

---

## [Author Response · Author response to Decision Letter 0]

26 Jul 2023

Journal Requirements:

All of the above comments have been incorporated in the revised Manuscript as well as in the Cover Letter. 

We provide an access # within the Manuscript.

This has been updated. 

This has been done and updated in the Manuscript.

6. Please upload a new copy of Figures 1 and 4 as the detail is not clear. Please follow the link for more information: https://blogs.plos.org/plos/2019/06/looking-good-tips-for-creating-your-plos-figures-graphics/

We now include revised, high-resolution figures 1 and 4 in the Manuscript.

7. We note you have included a table to which you do not refer in the text of your manuscript. Please ensure that you refer to Table 1 in your text; if accepted, production will need this reference to link the reader to the Table.

This has been done and updated in the Manuscript.

Reviewers’ Comments:

We are very thankful for the thoughtful comments of the reviewers. Please see our responses and revisions for each comment. 

Reviewer #1: This manuscript mainly reports the RNA-seq results of the periapical tissue of patients with periapical periodontitis, compares the gene expression differences between symptomatic and asymptomatic, or male and female, and uses RT-qPCR to verify key genes. There are however some concerns that need to be addressed satisfactorily by the authors before this manuscript could be considered acceptable.

1. About the Exclusion and Inclusion Criteria:

1) 12 patients were included for RNA-seq, however, I notice that the oldest was 89 years old and the youngest was 19 years old. I wonder if age is a factor that affects the RNA-seq results? The same concern applies to the race of the patients who were included.

We appreciate and concur with the reviewer's feedback regarding age and race as variables. We therefore, have conducted validation studies using RT-PCR on a selection of target genes. Our study includes a sample size of 6-10 individuals per group, encompassing a broad range of ages and both Hispanic and White for races. These additional experiments serve to support and validate the outcomes derived from the bulk RNA-seq studies. By incorporating diverse biological samples, we aim to minimize the impact of age- and race-related variability and strengthen the reliability of our findings. We have now included this point in the “Discussion” section of the manuscript. 

2) There is a lack of description of the details of each patient's condition, such as the degree and scope of periapical resorption in each patient, and whether the difference in the degree of infection will also affect the RNA-seq results?

We greatly appreciate the thoughtful comment provided. In our patient cohort study, we did not specifically evaluate the size of periapical bone resorption. It is important to note that this cohort of patients are defined by chronic non-healing periapical lesions that are excised during microsurgical procedures and therefore have chronic periapical pathosis. Additionally, no studies have established a direct link between the size or chronicity of periapical lesions and the pain phenotype, suggesting a lack of correlation. However, this warrants a future study. We have now included and clarified this point in the “Experimental

Procedures” section of the manuscript.

2. About the sample collection:

The author mentions “3-8 mm3 biopsies were collected” in the manuscript, but lacks the description of details, for example, were the biopsies taken from the apical foramen or from the center of the lesion? Or randomly take 3-8mm3 from the diseased tissue removed during the surgery?

Periapical lesions were collected in its entirety. We have specified this point in the “Experimental Procedures” section. However, patients that requested a pathology report had part of the lesion submitted to pathology for a biopsy report. 

3. About the RT-PCR:

1) In the section of Results, the authors mention that “Using biological samples (N=6-12) entirely different from the ones used in generating RNA sequencing datasets, we conducted validation studies using RT-PCR.” In the Methods section the authors do not mention these details.

We now provide demographic details for each sample used for RT-PCR validation studies in the “Experimental Procedures” section. 

2) Do the samples for PCR meet the same exclusion and inclusion criteria? I think the patient information for PCR should also be listed as a table in the methods section.

Yes, all patient samples for RT-PCR were collected using the same inclusion and exclusion criteria. We have now state this in the “Experimental Procedures” section.

4. About the Symptomatic or Asymptomatic patient:

1) The authors define whether symptoms are present through palpation and percussion, and patients with mechanical allodynia to palpation or percussion are considered symptomatic patients. Table 1 should also indicate whether patients with mechanical allodynia to both examinations or only one examination.

2) Were all examinations performed by the same person, and are the standards the same? The authors should add the description of the details of the examination.

Clinical exams were performed by multiple providers. However, each provider was calibrated prior to the start of the study with respect to clinical diagnosis and periapical lesion excision. Tissue procurement and storage was also standardized for all research personnel. We add this detail now in the “Experimental Procedures” section. 

Reviewer #2: This is a well conducted study, which is likely to have a great impact once published.

there are some minor comments: Fig 1-A and 1B and Fig 4: legends are not visible, please enlarge/bold for further clarity for the reader.

We apologize for the poor quality of figures for Fig1. and 4. We have now included new figures for Fig. 1 and 4. 

Please make sure all the abbreviations are defined at their first appearance in the manuscript.

Thank you and we now have proofread the manuscript for these.

---

## [Decision Letter · Decision Letter 1]

29 Aug 2023

PONE-D-23-12043R1Understanding Painful versus Non-Painful Dental Pain in Female and Male Patients: A Transcriptomic Analysis of Human BiopsiesPLOS ONE

Dear Dr. Ruparel,

Thank you for submitting your manuscript to PLOS ONE. After careful consideration, we feel that it has merit but does not fully meet PLOS ONE’s publication criteria as it currently stands. Therefore, we invite you to submit a revised version of the manuscript that addresses the points raised during the review process.

io to enhance the reproducibility of your results. Protocols.io assigns your protocol its own identifier (DOI) so that it can be cited independently in the future. For instructions see: https://journals.plos.org/plosone/s/submission-guidelines#loc-laboratory-protocols. Additionally, PLOS ONE offers an option for publishing peer-reviewed Lab Protocol articles, which describe protocols hosted on protocols.io. Read more information on sharing protocols at https://plos.org/protocols?utm_medium=editorial-email&utm_source=authorletters&utm_campaign=protocols.

We look forward to receiving your revised manuscript.

Kind regards,

Artak Heboyan, Ph.D.

Academic Editor

PLOS ONE

Reviewers' comments:

Reviewer's Responses to Questions

**Comments to the Author**

1. If the authors have adequately addressed your comments raised in a previous round of review and you feel that this manuscript is now acceptable for publication, you may indicate that here to bypass the “Comments to the Author” section, enter your conflict of interest statement in the “Confidential to Editor” section, and submit your "Accept" recommendation.

Reviewer #3: (No Response)

2. Is the manuscript technically sound, and do the data support the conclusions?

Reviewer #3: (No Response)

3. Has the statistical analysis been performed appropriately and rigorously? 

Reviewer #3: Yes

4. Have the authors made all data underlying the findings in their manuscript fully available?

Reviewer #3: (No Response)

5. Is the manuscript presented in an intelligible fashion and written in standard English?

Reviewer #3: Yes

6. Review Comments to the Author

Reviewer #3: this is a well written paper. my main concerns are:

1. In the main analysis of transcriptional changes all of the symptomatic males were Hispanic and the asymptomatic males were Asian and White. my concern is that the results could be due to population stratification rather than true differences in expression changes.

2. in Table 2 there is no information on Asymptomatic females. Were they not included in the secondary analyses? could you please clarify?

7. PLOS authors have the option to publish the peer review history of their article (what does this mean?). If published, this will include your full peer review and any attached files.

Reviewer #3: No

---

## [Author Response · Author response to Decision Letter 1]

30 Aug 2023

We thank the reviewer for the positive feedback and questions. Please see our responses in blue font. 

Reviewer #3: this is a well written paper. my main concerns are:

1. In the main analysis of transcriptional changes all of the symptomatic males were Hispanic and the asymptomatic males were Asian and White. my concern is that the results could be due to population stratification rather than true differences in expression changes.

We appreciate this concern of the reviewer. While our initial sample cohort might appear limited in terms of racial diversity within the symptomatic male (SM) group, our samples encompassed both Hispanic and White male populations. This broad representation ensures that our findings are not confined to a singular demographic, thus enhancing the applicability and relevance of our results across different populations. This deliberate inclusivity was aimed at preventing any undue bias or overrepresentation and promoting a more comprehensive understanding of the underlying biological processes. Collectively, we have diligently taken steps to enhance the robustness and validity of our outcomes.

In our revised manuscript, we have acknowledged this within the “Discussion” section of the manuscript. It states as follows. 

“A small sample size of N=3 with a wide age range and race was used for bulk RNA sequencing, which potentially reduces the generalizability of our findings. Moreover, the majority of patients declined a histopathologic assessment of the biopsy samples. Given the incidence of periapical granuloma vs. periapical abscess vs. periapical cyst, the differential cellular compositions of each of these diagnoses present a biological confound in our patient samples. However, we conducted validation studies using RT-PCR of several target genes with separate biological samples with an N=6-10 per group. This greatly increases the rigor and internal validity of our findings and minimize the impact of age, race or histopathology-related variability.”

2. In Table 2 there is no information on Asymptomatic females. Were they not included in the secondary analyses? could you please clarify?

We appreciate this question. In our validation studies, we opted for the "Convenience Sampling" method to streamline the process. This approach involved focusing on genes with a robust expression level, indicated by an RPKM (Reads Per Kilobase Million) value exceeding 10, as this threshold ensured a reliable detectable range when employing the RT-PCR technique. The symptomatic male (SM), symptomatic female (SF), and asymptomatic male (AM) groups exhibited a higher prevalence of genes meeting the RPKM criterion, making them conducive for efficient validation through RT-PCR. The accessibility and availability of samples (N=6-12) from these groups were also comparably greater in comparison to the asymptomatic female (AF) group. This difference in accessibility contributed to the expeditious validation of our bulk RNA sequencing findings. However, we acknowledge the need for future studies to encompass a a broader spectrum of genes, enhancing the comprehensiveness and reliability of our results as well as a crucial precursor to subsequent functional studies.

---

## [Editor Report · Decision Letter 2]

6 Sep 2023

Understanding Painful versus Non-Painful Dental Pain in Female and Male Patients: A Transcriptomic Analysis of Human Biopsies

PONE-D-23-12043R2

Dear Dr. Ruparel,

We’re pleased to inform you that your manuscript has been judged scientifically suitable for publication and will be formally accepted for publication once it meets all outstanding technical requirements.

Kind regards,

Artak Heboyan, Ph.D.

Academic Editor

PLOS ONE
---

## [Editor Report · Acceptance letter]

12 Sep 2023

PONE-D-23-12043R2 

Understanding Painful versus Non-Painful Dental Pain in Female and Male Patients: A Transcriptomic Analysis of Human Biopsies 

Dear Dr. Ruparel:

I'm pleased to inform you that your manuscript has been deemed suitable for publication in PLOS ONE. Congratulations! Your manuscript is now with our production department. 

Kind regards, 

on behalf of

Dr. Artak Heboyan 

Academic Editor

PLOS ONE